# Usability Evaluation and Classification of mHealth Applications for Type 2 Diabetes Mellitus Using MARS and ID3 Algorithm

**DOI:** 10.3390/ijerph19126999

**Published:** 2022-06-08

**Authors:** Kamaldeep Gupta, Sharmistha Roy, Ayman Altameem, Raghvendra Kumar, Abdul Khader Jilani Saudagar, Ramesh Chandra Poonia

**Affiliations:** 1Faculty of Computing and Information Technology, Usha Martin University, Ranchi 835103, India; kamal.sxcranchi@gmail.com (K.G.); sharmistha@umu.ac.in (S.R.); 2Department of Computer Science and Engineering, College of Applied Studies and Community Services, King Saud University, Riyadh 11533, Saudi Arabia; aaltameem@ksu.edu.sa; 3Department of Computer Science and Engineering, GIET University, Rayagada 765022, India; raghvendra@giet.edu; 4Information Systems Department, Imam Mohammad Ibn Saud Islamic University (IMSIU), Riyadh 11432, Saudi Arabia; 5Department of Computer Science, CHRIST (Deemed to be University), Bangalore 560029, India; rameshchandra.poonia@christuniversity.in

**Keywords:** MARS, ID3, mHealth applications, T2DM, usability, decision making

## Abstract

The rapid growth of mHealth applications for Type 2 Diabetes Mellitus (T2DM) patients’ self-management has motivated the evaluation of these applications from both the usability and user point of view. The objective of this study was to identify mHealth applications that focus on T2DM from the Android store and rate them from the usability perspective using the MARS tool. Additionally, a classification of these mHealth applications was conducted using the ID3 algorithm to identify the most preferred application. The usability of the applications was assessed by two experts using MARS. A total of 11 mHealth applications were identified from the initial search, which fulfilled our inclusion criteria. The usability of the applications was rated using the MARS scale, from 1 (inadequate) to 5 (excellent). The Functionality (3.23) and Aesthetics (3.22) attributes had the highest score, whereas Information (3.1) had the lowest score. Among the 11 applications, “mySugr” had the highest average MARS score for both Application Quality (4.1/5) as well as Application Subjective Quality (4.5/5). Moreover, from the classification conducted using the ID3 algorithm, it was observed that 6 out of 11 mHealth applications were preferred for the self-management of T2DM.

## 1. Introduction

Millions of people’s lives have been transformed by mobile technology [1]. Smartphones offer the perfect balance of user-friendliness and high-functioning customizable content. The accelerated development of several mHealth applications has resulted from the increased use of smartphones. The term “mHealth” relates to services associated with clinical and public healthcare that are accessible via smartphone devices and provide health-related data and activities to people from anywhere and at any time [2]. Users of mHealth applications are encouraged to participate in their own healthcare management strategy, especially when it comes to the prevention and/or self-management of chronic conditions [3]. The term mHealth is considered as the use of mobile technology to enhance health care performance and effectiveness. It has emerged as a viable approach for diabetic patients to enhance self-management and healthcare-related performances [4,5,6]. These mHealth applications make it possible for users to stay in touch with healthcare professionals in ways that were not possible before [7]. The rising frequency of T2DM (Type 2 diabetes mellitus) and the growing severity of management programs are putting pressure on health systems, particularly in primary care, where doctors often do not have enough time for diabetic patients. Smartphones and wearable devices offer novel ways for managing T2DM that are extremely scalable. There is a necessity for the qualitative assessment of these T2DMapplications as the number of health applications and their advantages for T2DM patients are expanding. Users frequently evaluate an application’s quality on the basis of its description, star rating system, or remarks, but all these criteria are not always true, and applications might not be medically appropriate or may constitute a security concern [8,9].

To analyze the effectiveness of the prominent free applications for T2DM management, we utilized the Mobile App Rating Scale (MARS) [10], a robust and recognized rating methodology for evaluating the quality of mHealth applications. MARS is a straightforward, simple, and consistent method for categorizing and evaluating the effectiveness of mHealth applications [11]. It may be used as a checklist to develop and design high-quality mHealth applications [10].

Data mining is the process of uncovering unknown patterns in large datasets that are possibly valuable and eventually understood [12]. Using a combination of statistics, machine learning, and database systems, data mining aims to retrieve usable information from large data sets and convert it into an intelligible structure. Algorithms related to data mining employ trained datasets to create a framework that may be utilized to forecast hidden or unknown data. Classification is a technique for predicting a data instance’s category class label and for classifying it under one of the specified classes. Classification is a two-stage process in which a classification algorithm employs a training dataset to create a classifier, which is further utilized to forecast the class label of an associated unlabeled data instance in the second phase. The classifier functions as a mapping between a data instance and a label. The decision tree is among the most widely used classifiers. In this study, we used an ID3 algorithm to make a decision depending on datasets and multiple evaluation criteria to determine whether the application is preferable or not.

The primary goal of this study was to perform a systematic assessment of the 11 mHealth applications to assist in the self-management of T2DM. This article presents an overview of these 11 mHealth applications related to the self-management of T2DM. The idea was to use the MARS tool to analyze and evaluate T2DM applications in order to find and present the optimal ones to users. In addition, the ID3 (Iterative Dichotomiser 3) algorithm was utilized to create a decision tree which predicts whether or not T2DM mHealth applications are preferable. The most important functions and features of these applications in terms of assisting diabetics were also explored.

Therefore, the purpose of this study was to collect data on the most widely used T2DM mHealth applications (as indicated in Table 1) and assess their potential quality as an aid for T2D prevention, with a focus on Indian users. Table 1 lists the 11 applications that were downloaded and reviewed for this study.

The objectives of the study are mentioned below:Evaluating the usability of mHealth applications for T2DM and ranking the applications using MARS methodology.Choosing the best mHealth applications for T2DM using the ID3 decision making algorithm.

The remaining parts of this work are organized as follows. Section 2 presents proposed methodologies for identifying the best mHealth applications based on usability. Section 3 discusses the papers that were reviewed for the MARS and ID3 algorithm methodologies for usability evaluation. Section 4 details how the methodologies employed in this work were implemented. Section 5 shows the analyses and validates the results. Section 6 focuses on the discussion, and lastly, Section 7 presents the conclusion.

## 2. Proposed Methodology

### 2.1. Alternatives Used for the Study

We considered eleven T2DM mHealth applications (considered as alternatives) for the evaluation of usability and ranking, as well as classified them based on users’ feedback; the applications are those listed in Table 1.

**Glucose Buddy (Alt1):** This application provides the following services:It can track blood sugar, insulin, weight, blood pressure, A1C, and other trends, as well as record blood glucose, medicine, and meals all in one entry;Adds notes to entries for future reference and automatically track walks and other aerobic exercises;It offers real-time blood sugar monitoring as a straightforward and convenient way to control diabetes;It provides expert advice and help.

**Diabetes: M (Alt 3):** By offering the following services, this mHealth application delivers everything necessary for optimal health management:It assists in presenting detailed information to the user. It provides good, remote diabetes management;It presents the material in a statistical format (such as a bar chart) to aid comprehension;It can recognize trends and look for any pre-defined recurring issues as well as the causes of their occurrence;It has an insulin bolus calculator that calculates insulin based on dietary information.

**mySugr (Alt 2):** The following are the best features of the mySugr application:It may record blood glucose, medicine details, and meals all in one entry, as well as track blood sugar, insulin, weight, blood pressure, A1C, and other trends;It has a customizable logging screen that can record data from a Bluetooth-enabled blood glucose meter and analyze the trend to give you a rapid summary of your blood glucose levels;It has a superior search capability for documenting meals and activities, which makes diabetes management easier;Reminders and recommendations on blood pressure, diabetes, food, exercise, foot, eyes, kidneys, and cardiovascular risks are available;It is capable of providing the highest level of data protection, as required by the general data protection rule (GDPR);Users can get important medical data from the app and export it as a PDF report or an Excel file.

**BeatO Smart Diabetes Management:** Some popular features are:It may record blood glucose, medicine details, and meals all in one entry, as well as track blood sugar, insulin, weight, blood pressure, A1C, and other trends;All blood glucose readings are instantly saved on the app’s blood sugar diary and may be evaluated using simple graphs;It uses color coding to show high, low, and normal blood sugar levels;It syncs with the fitness tracker so you can track your steps and see how many calories you burned in a day right on the app. It works with Google Fit, Apple Health Kit, Fit Bit, and other popular fitness trackers to provide a unified view of health and activity data.

**Blood Glucose Tracker (Alt 4):** The following are the services provided by this application:It measures blood glucose at various levels throughout the day (for example, at breakfast, lunch, and dinner) to help patients maintain efficient blood sugar control;Among other things, it can monitor blood pressure, weight, and HbA1c levels;It allows the user to filter the history by event type/tag, which is useful for keeping track of things such as exercise reactions and food preferences.

**Health2Sync-Diabetes Care:** The following are the characteristics of this mHealth application:It conveniently logs blood glucose readings and allows users to add notes, pick medications, eat foods, and attach images;It keeps track of blood pressure measurements and allows you to enter systolic, diastolic, and pulse readings to monitor how you are doing with blood pressure control;It can keep track of weight changes and one can enter the weight and body fat, set goals, and easily see the progress made;It has a dashboard that gives a quick analysis of the recent blood sugar history in table and graph formats. One can see whether the readings are within range, the movement of the readings, and can even chart differences between before-meal and after-meal readings;It has a diary that allows the user to easily review past blood sugar records. One can quickly see what factors contributed to a high or low reading;It can keep track of the A1C history and allows the data entered on the app to be exported as a PDF Report or Excel file.

**OneTouch Reveal (Alt 5):** The following are the distinguishing characteristics of this mHealth application:It organizes blood sugar information in a way that beginner users may understand using a unique color-coding technique;It sends out automatic alerts when repeated highs or lows occur, allowing the user to take appropriate action;It sets a daily goal for the logging of steps, carbs, and activities;It reminds the user when it is time to take a blood sugar test and when to take insulin.

**Diabetes Diary-Blood Glucose Tracker:** The following are the features of Intellin Diabetes Management:It keeps track of blood glucose levels and blood pressure readings so one can monitor how one is doing with blood pressure control;It can keep track of weight changes, and one can enter the weight and body fat, set goals, and easily see the progress made;It can aid in the comparison of physical activity and weight-checking findings entered in various sessions;It can give patients specific advice on how to effectively handle a problem;It reminds patients when to take medicine and about other self-care activities, as well as predicts their condition based on data entered into the system.

**Diabetes Forum:** Some well-known features of this app are as follows:It keeps track of the blood glucose levels, blood pressure, activity, BMI, and more with ease;It can assist with weight management and HbA1c levels;It can aid in the management of Type 1, Type 2, and gestational diabetes by providing motivating patterns and feedback;It can be used to keep track of medications and data can be exported as a PDF report or an Excel spreadsheet.

**Intellin Diabetes Management:** The following are the features of Intellin Diabetes Management:It allows the user to easily track blood pressure, activity, BMI, blood glucose levels, and more;It enables the monitoring and management of blood sugar levels with the blood sugar monitor;It allows one to discover the top diabetic risk factors and how to manage them if you have diabetes;It provides reminders and recommendations on how to manage the blood pressure, glucose, nutrition, activity, foot, eyes, kidneys, and cardiovascular risks;It uses motivational trends and feedback to help manage Type 1, Type 2, and gestational diabetes;It provides diabetes analysis on a daily, weekly, monthly, and annual basis;It allows the user to backup diabetic data safely;Over 100 devices can be connected to track the activity, weight reduction, BMI, blood pressure, blood sugar, and other metrics. With the addition of other devices, one will be able to take control of their diabetes quickly.

**Diabetes Connect:** Some standard features of Diabetes Connect are as follows:It enables users to add notes, choose medications, eat foods, and upload photographs to their blood glucose readings;It can keep track of weight changes and allows users to enter their weight and body fat percentage, set goals, and track their progress;It can also track blood pressure and HbA1c levels, among other things;It establishes a daily goal for tracking steps, carbohydrates, and activities;It allows users to get important medical information and export data as a PDF report or an Excel spreadsheet from the app.

### 2.2. PRISMA for Search Strategy and Inclusion and Exclusion Criteria

PRISMA was used to carry out a systematic search. PRISMA is often successfully employed in the search for applications for various research in the health field as it is a widely used technique. PRISMA is effective for providing reviews of different types of studies because of its broad applicability [13]. PRISMA is a publication guide that has a flowchart [14] consisting of four phases, which promotes transparency and consistency in reporting related to systematic reviews. Identification is the first phase, followed by Screening, Eligibility, and Included.

### 2.3. Mobile App Rating Scale (MARS)

MARS, established by Stoyanov et al. [15], can be utilized to evaluate or analyze the effectiveness of mHealth applications. MARS is a methodology for assessing the effectiveness of medical applications and comprises five subscales: engagement, functionality, aesthetics, and information quality, along with the application’s subjective quality [10,15]. Application Quality (Category I), which includes Sections A–D, and the Application Subjective Quality (Category II), which involves Section E, are the two key categories in MARS. The four sections of application quality requirements include engagement (ENGT), functionality (FUNT), aesthetics (AEST), and information quality (INFN). As indicated in Figure 1, these sections are even further divided into 19 sub-sections. Each of the 19 sub-sections is rated using a 5-point Likert scale (1-inadequate, 2-poor, 3-acceptable, 4-good, 5-excellent). Depending on this, a mean quality score is presented for each of the sections from A to D. Averaging the mean scores yields an aggregate mean score for such categories, which is used to determine the application quality (AQ) mean score. In addition, questions in Section E about recommending the application, frequency of using the application, willingness of the user to buy the application, and the aggregate rating given to the application as per the user’s perspective are included in the category “Application Subjective Quality (ASQ)”. The applications are rated and presented in several categories.

### 2.4. Decision Tree Classifier

Decision tree is amongst the most efficient classification techniques in the domains of pattern recognition and data mining, and it is not just associated with databases, artificial intelligence, and other disciplines, but also has tremendous theoretical research relevance. One of the recursive divide and conquer techniques is the greedy top-down decision tree [16] learning algorithm. A decision tree resembles a tree-like structure. The internal nodes are labelled using attributes and represent a test on that attribute, outgoing edges represent the test’s conclusion, and leaf nodes represent classes. The training dataset is divided into two or more subgroups, depending on the values of the attributes labelled on the nodes. A decision tree is made up of nodes (rectangular boxes) and edges (arrows) that are created using a dataset, which is a table with columns that indicate features/attributes and rows that correspond to records. Every node is utilized to make a decision (regarded as the decision node) or to depict an outcome (regarded as the leaf node). An attribute selection measure is utilized to choose attributes at each stage. It is a heuristic for determining which splitting criterion separates the training dataset into different classes. ID3, in which the attribute selection measure is the Information Gain, is one of the most often used decision tree algorithms. The algorithm of the decision tree operates on subsets of data and remainder attributes in a recursive manner. The recursion ends when any one of the following terminating conditions is true: leaf nodes are labelled with that class if all tuples of that class are acquired; if the attribute list is empty, the leaf node is named using the dominant class of tuples; or the leaf node is named using the dominant class of the node’s parent training tuples if there are no more tuples left.

### 2.5. ID3 Algorithm

J. Ross Quinlan of the University of Sydney created ID3 [17]. ID3 was published for the first time in 1975 in Machine Learning [17], vol. 1, no. 1. At each phase of the ID3 method, attributes are dichotomized (divided) into two or more groups iteratively (repeatedly) [18].

ID3 [19] is basically a supervised learning technique that uses a set of samples to generate a decision tree. Subsequent samples are categorized using the resulting tree. The ID3 algorithm creates a tree depending on the information gain acquired through the training cases, which is then utilized to categorize test data. For categorization, the ID3 algorithm typically utilizes nominal attributes with no missing data [19].

To construct a decision tree, ID3 employs a top-down greedy strategy. Briefly stated, the top-down technique means we build the tree from the top down, but the greedy approach involves choosing the optimal feature at the time in order to generate a node at each iteration. ID3 is frequently utilized for tasks that utilize nominal features in categorization. The ID3 algorithm chooses the optimal feature at each step of the decision tree development.

ID3 Steps:Determine each feature’s Information Gain.Divide dataset S into subsets utilizing the attribute that has the highest Information Gain, assuming that not all rows correspond with the similar class.Create a decision tree node with the feature that gives you the highest information.If all corresponds to the similar class, create the present node with a leaf node that has the class as its label.Continue until the decision tree is completely filled with leaf nodes or until you run out of attributes.

The ID3 algorithm chooses the attribute to be divided on the basis of two metrics:

**(1) Entropy (ENT) Metric:** The dataset’s entropy is a measurement of the disorder in the dataset’s target attribute. It determines how much information is contained in a specified attribute. For the remaining attributes, entropy is determined. The attribute with the lowest entropy is split.

When all entries inside the target column are homogenous (similar), ENT is 0; when the target column includes an identical number of entries from both classes in binary classification (when the target column has just two types of classes), ENT is 1. 

Our dataset is denoted by DS, and the ENT is determined as follows:ENT(S) = −∑ prk × log_2_(prk); k = 1 to n
where n denotes the number of classes present in the target column, prk is the probability of class k and is calculated as the ratio of “amount of rows having class k inside the target column” to “total amount of rows” that belong to the dataset.

**(2) Information Gain (IG):** IG is an attribute selection metric that assesses how the feature distinguishes or categorizes target classes and evaluates the reduction in entropy. Depending on the entropy, the option with the maximum IG (information which is the most valuable for classification) is chosen. It also reflects the average amount of information needed to classify each tuple inside the dataset.

IG associated with a feature column A is determined as follows:IG(DS, A) = ENT(DS) − ∑ ((|Sv|/|S|) × ENT(DSv))
where the set of rows in DS in which the feature column A has the value v is denoted by DSv, |DSv| denotes the amount of rows in DSv, and |DS| denotes the amount of rows in DS. 

## 3. Literature Review

### 3.1. Review Works on MARS

Stoyanov et al. [10] established a multidimensional rating scale to classify and rate the effectiveness of mHealth applications, and it is both dependable and accurate. A search of the literature was performed in order to find articles with specific quality rating criteria related to the web or applications. An expert panel analyzed the existing criteria for app quality assessment in order to create the new MARS subscales, items, classifiers, descriptors, and anchors.

In China, Gong et al. [20] systematically reviewed and assessed diabetic self-management mobile applications. The particular goals were to (1) present an outline of the various Chinese mobile applications involved in the self-management of diabetes, (2) assess the effectiveness of these applications using standardized rating scales, and (3) characterize the essential features of the applications in assisting diabetics.

Stec et al. [21] assessed the MARS instrument and its use by physicians, as well as a number of primary health care and wellness applications which have been evaluated with this tool. With the help of the MARS tool, the authors evaluated 23 medical applications.

Escriche-Escuder et al. [22] analyzed the mHealth applications available for managing low back pain and utilized MARS to assess their effectiveness and present a summary of their characteristics, quality, and operability. In September 2019, two independent reviewers investigated the official Android (Play Store) and iOS (Apple Store) shops for localization in Spain and the United Kingdom, looking for applications linked to low back pain therapies. In the end, seventeen applications were included. MARS was utilized for assessing the app’s quality. 

Bardus et al. [23] analyzed the efficiency and information related to the prominent weight management applications on iTunes and Google Play and subsequently described their features. To get a more comprehensive assessment as compared to app store user ratings, the MARS scale was employed to evaluate engagement, functionality, aesthetics, and information quality. Only weight management applications were selected after two researchers examined the descriptions. The MARS and earlier specified categories of strategies essential to behavioral changes were used to independently assess features, app quality, and content.

Salazar et al. [24] utilized the App Store and Play Store to collect 18 pain-related mobile applications. The MARS tool was utilized to assess their quality. Every app’s scores (for each part and the total score) were recorded, and the mean score as well as the standard deviation were reported, thus providing a complete picture of the app’s efficiency. Based on the regularity of the distribution (Shapiro–Wilk test), the section scores were compared between the groups specified by the tertiles using the Kruskal–Wallis test or an analysis of variance (ANOVA).

Grainger et al. [8] evaluated the characteristics and reliability of applications to help people supervise the disease activity related to Rheumatoid arthritis (RA) by (1) providing an overview of the available applications, especially the equipment utilized for measuring disease activity associated with RA; (2) correlating app characteristics with regulations of the European League against Rheumatism (EULAR) and the American College of Rheumatology (ACR) in order to track disease activity associated with RA; and (3) scoring the application quality by applying MARS.

Salehinejad et al. [25] devised a valid method for rating and assessing the quality of COVID-19 mHealth applications, with the goal of creating a framework for the future development of the mHealth app. We identified applications for the iOS and Android platforms using COVID-related terms in this investigation. Thirteen applications were chosen for the purpose of reviewing. Two reviewers independently evaluated the quality of the applications using MARS. According to this study, many COVID-related applications meet reasonable performance, information, or operability criteria, but they should emphasize aesthetic and fascinating elements to increase the overall quality and be appreciated by users.

Knitza et al. [26] compiled a list of mHealth applications related to rheumatology that are present in German app stores, utilizing MARS to assess app quality; they then compiled succinct, readily available summaries for rheumatologists and patients. German rheumatology mobile applications for patients and physicians were found through a systematic search strategy of the German Application Store as well as the Google Play store. Eight physicians, four utilizing Android smartphones and four with iOS smartphones, employed MARS to independently rate app quality.

Sullivan et al. [27] discovered, defined, and graded the quality of currently available smartphone applications that track personal travel and nutritional behavior while also calculating the carbon cost and associated health repercussions.

Santo et al. [28] assessed the functionality and efficiency of applications that give medication reminders and are accessible from the Australian iTunes and Google Play store in order to find high-quality applications. This study used a step-by-step approach, which included (1) a strategy for selection; (2) an assessment of eligibility; (3) a process for app selection; (4) the extraction of data involving a predefined feature set; (5) the performance of an examination by categorizing the applications into fundamental and advanced applications that provide medication reminders; and (6) a quality evaluation with the help of a reliable tool known as MARS for evaluating mHealth applications.

Creber et al. [29] assessed commercial applications to find and evaluate the features of patients who make use of mHealth applications that enable heart failure symptom management. MARS, the IMS Institute for Healthcare Informatics performance rating scores, and the regulations of the Heart Failure Society of America (HFSA) for non-pharmacologic therapy were used to evaluate applications that met the inclusion requirements. A group of 2–4 reviewers installed and rated applications independently, determining inter-class correlations among reviewers and reaching a consensus through conversation.

Larco et al. [30] defined and reviewed the efficacy of educational applications for people suffering from Down syndrome, autism, and cerebral palsy. In the Apple App Store, a thorough search was undertaken. A panel of evaluators utilized MARS—which consisted of twenty-three elements divided into subscales that include engagement, functionality, aesthetics, information, and subjective quality—to analyze and rate the quality and usefulness of 50 applications.

Moseley et al. [31] assessed and ranked smartphone applications present in the Apple iPhone and Android Play Stores that aimed to enhance eating behavioral patterns by addressing the habit-forming element of overeating or the consumption of unhealthy foods, including sugar addiction.

### 3.2. Review Works on ID3 Algorithm

In their study, Kale et al. [32] discussed how they used the ID3 algorithm to perform automatic menu planning selection for children, as indicated by a nutritional management system. This study was conducted with the help of an Indian food database since many Indian children suffer from malnutrition as a result of their mothers’ lack of knowledge about nutrition. The key challenge in the implementation was how to propose a certain food item from the food database according to certain characteristics such as likelihood, availability, nutritional content, and the child’s decision. The result would then aid in the selection of foods from the database, ensuring that a deficiency would not arise in the near future and that the child would receive an appropriate diet plan. To make the best choice among the various foods, the ID3 algorithm was applied.

Aalagadda et al. [33] recognized appropriate attributes from socio-demographic, academic, and institutional data for first-year university students in the form of a model machine learning tool which automatically specifies whether the student can proceed with his studies or use the classification methodology based on the ID3 algorithm, which is a widely used decision tree.

To construct a decision tree, Surya et al. [34] employed the ID3 technique. ID3 constructs a decision tree using entropy and information gain. This study provides an overview of ID3’s use in a variety of sectors, including medicine, health, education, computer forensics, web attacks, and food databases.

Adhatrao et al. [35] created a system that could forecast student performance based on their previous performance using classification data mining methodologies. They examined a data set that included information about students such as gender, grades in the 10th and 12th board examinations, marks and positions in entrance exams, and the first-year outcomes from the senior students. On this data set, the overall and individual efficiency of newly enrolled students in the coming examinations were predicted using the ID3 and C4.5 classification algorithms.

To implement data classification, Zhang et al. [36] utilized a standard data set as the original discrete experimental data and evaluated the entropy and information gain of each attribute of the data. The attributes related to the information gain that minimizes the largest entropy were chosen as the ideal classification attribute for the development of the decision tree after traversing the structure of the tree.

Hazra et al. [37] used relevant attributes involving the quantitative and qualitative features of a job candidate’s experience, work status, present salary, level of education, whether from a top-tier institution or not, and internships to create a model for predicting a candidate’s hiring. The ID3 method was used in this research to produce decision rules that could be utilized to forecast the likelihood of hiring a job candidate. This aids the recruiter in quickly deciding whether or not to hire an applicant and in selecting the best candidate for the job.

## 4. Implementation of MARS and ID3 Algorithm

### 4.1. Ranking of mHealth Applications Based on Usability Using MARS

#### 4.1.1. Search Strategy and Inclusion and Exclusion Criteria

As described in Section 3.1, a systematic search was conducted in accordance with PRISMA. Furthermore, two separate primary reviewers explored Google Play (Android Platform) applications using terms or keywords such as type 2 diabetes, blood glucose diabetic control, healthy living, and fitness. The applications had to be in English, be free, be related to T2DM patients, have over 10,000 downloads, and not require a subscription. Applications were taken into consideration if they were utilized for the self-management of diabetes and featured at least one of the following features: monitoring of blood glucose, management of nutritional and physical activity, medication administration, and aided in the avoidance of diabetes-related complications. Paid applications, non-English language applications, non-T2DM-specific applications, and duplicated applications were all excluded from the Identification phase when employing PRISMA. The exclusion criteria throughout the screening phase were irrelevant content to T2DM and the need for registration. The exclusion criteria for the Eligibility phase were insufficient information and no longer working. The rest of the applications were downloaded and analyzed with MARS during the included phase, and a decision tree was created with the help of the ID3 algorithm. Two reviewers independently reviewed, rated, and evaluated each of the eleven applications.

#### 4.1.2. Quality Assessment of the Applications Using MARS

MARS scoring was completed separately by two reviewers, one who has a master’s degree in health information management and presently working as research consultant in a medical firm, particularly on diabetes, and the other with master’s degree in medical informatics and has been working as a dietitian for several years. The two primary reviewers downloaded the 11 shortlisted applications shown in Table 1 to their Android phones and utilized them thoroughly to perform a complete assessment before evaluating them using MARS. The MARS scale was employed to construct a template for the extraction of the data. The template’s first sheet comprised details of the application, application quality ratings were incorporated in the second sheet, the third sheet included subjective application quality, and the last sheet provided a MARS subscale summary. While scoring each application, the reviewers utilized all of the application’s features to acquaint themselves with it. Both the researchers received training on T2DM and how to use the templates to perform MARS. The best quality applications were identified using the mean MARS scores.

The MARS methodology was used to calculate the mean value of all application aspects, including engagement, functionality, aesthetics, and information. In this study, all of the T2DM mHealth applications were evaluated using this method. Application Quality Scores for all of the applications were also determined. The mean values of each attribute and the overall Application Quality mean score associated with all the applications are shown in Table 2. Furthermore, subjective quality rating scores for all applications were calculated based on factors such as Application Recommendation, Frequency of Using the Application, Willingness of the users to Buy the Application, and Aggregate Rating of the Application. The Application Subjective Quality mean score for all applications is shown in Table 3.

#### 4.1.3. Data Analysis

The general information associated with the application and the outcomes of the MARS tool as utilized by the reviewers were first loaded onto Excel before making use of the SPSS software (IBM, New York, USA) to evaluate the results. Following that, descriptive statistics were utilized to present general information related to the applications as well as the outcomes of the MARS tool’s reviews. Application quality mean scores for all the 11 applications were identified. Subjective Quality mean scores of all the applications were determined. The user ratings found in application stores, the amount of downloads, the quality factors described by the MARS score, and the subjective quality mean score were all analyzed using Spearman’s correlation analysis.

### 4.2. Selection of mHealth Applications Based on Usability Using ID3 Algorithm

#### Feature Selection

In this work, we demonstrate the use of the ID3 decision tree to classify and predict application selection with qualifiers for engagement such as bad, average, and good; for functionality such as low, moderate, and high; for aesthetics such as ugly, normal, and attractive; for information such as weak, medium, and strong, and for the necessary conditions such as preferred or not preferred. Table 4 shows the values and the mean score range assigned to the attributes ENGT, FUNT, AEST, and INFN, depending on the mean score generated for the attributes, by using the MARS methodology. Table 5 provides a summary of decision-making factors or necessary conditions that can lead to whether the applications would be preferred or not. 

The target class “App selection”, as shown in Table 5, is based on the application’s specific quality score calculated with the help of MARS. If the value of the application specific quality score is greater than 3, the target class value is “Yes”, which means that the application is preferable, and the value is “No” if the application is not preferable.

The decision column contains 11 instances, with two possible outcomes: “YES” or “NO” for preferring or not preferring the applications, respectively. Six selections are labelled “YES” (preferred), while the other five are labelled “NO” (not preferred).

As previously mentioned, the application’s features include engagement, functionality, aesthetics, and information. They can include the following values:

Engagement = {Bad, Average, Good}

Functionality = {Low, Moderate, High}

Aesthetics = {Ugly, Normal, Attractive}

Information = {Weak, Medium, Strong}

We must identify the attribute which will serve as the decision tree’s root node. For the four qualities, entropy and information gain are computed.


**
*Entropy and Information Gain*
**


The entropy of the complete dataset S:**ENT(DS)** = −6/11 × log2(6/11) − 5/11 × log2(5/11) = 0.99


**
*Note:*
**
*The entropy is 0 when the entries specified inside the target column are the same (meaning that they have no randomness).*


**1.** 
**Determination of the Entropy and Information Gain for the first attribute—Engagement**


|DS| = 11. The value of v can be bad, average, or good

ENT (DSbad) = −0/2 × log2 (0/2) − 2/2 × log2 (2/2) = 0

ENT (DSaverage) = −3/6 × log2 (3/6) − 3/6 × log2 (3/6) = 1

ENT (DSgood) = −3/3 × log2 (3/3) − 0/3 × log2 (0/3) = 0


**IG Calculation**
IG(DS, Engagement) = ENT(S) − 2/11 × ENT(DSbad) − 6/11 × ENT(DSaverage) − 3/11 × ENT(DSgood) 
= 0.99 − 2/11 × 0 − 6/11 × 1 − 3/11 × 0


IG(DS, Engagement) = 0.44

**2.** 
**Determination of the Entropy and Information Gain for the second attribute—Functionality**


|DS| = 11. The value of v can be low, moderate, or high

ENT (DSlow) = −0/4 × log2 (0/4) − 4/4 × log2 (4/4) = 0

ENT (DSmoderate) = −2/3 × log2 (2/3) − 1/3 × log2 (1/3) = 0.92

ENT (DShigh) = −4/4 × log2(4/4) − 0/4 × log2(0/4) = 0


**IG Calculation**
IG(DS, Functionality) = ENT(DS) − 4/11 × ENT(DSlow) − 3/11 × ENT(DSmoderate) − 4/11 × ENT(DShigh) 
= 0.99 − 4/11 × 0 − 3/11 × 0.92 − 4/11 × 0


IG(DS, Functionality) = 0.74

Similarly, the information gain for Aesthetics and Information are as follows:

IG(DS, Aesthetics) = 0.3

IG(DS, Information) = 0.44

As the information gain of the functionality attribute is the highest, the splitting will be executed based on Functionality, and it is applied as a decision attribute onto the tree’s root node (Figure 2).

As there are three possible values for Functionality, the root node contains three branches (Low, Moderate, and High).

## 5. Results

### 5.1. Systematic Search and Screening

From the first Google Play search, 74 applications associated with T2DM were identified. Following an examination of the title and description of the application, it was found that the majority of the applications were irrelevant to T2DM, were not in English, or were paid and duplicated applications. According to the raters, these applications did not match the inclusion criteria set for the self-management of diabetes. As a result, 31 applications were excluded in this phase, leaving 43 applications for the next step. Now, during the screening phase, it was discovered that 19 applications had T2D-unrelated information and required registration.

As a result, 24 applications were approved and were subsequently downloaded and analyzed; however, 13 applications could not be analyzed for the following reasons: (1) they lacked sufficient information, (2) they were no longer available, (3) they were no longer operating, or (4) the download failed. The flow diagram (Figure 3) depicts the selection process as well as the exclusion categories. Eventually, eleven applications for the Android platform were chosen for analysis.

Accordingly, those 11 applications were included in the MARS evaluation. The MARS score was calculated using the attributes ENGT, FUNT, AEST, INFN, and subjective quality. The outcome of the systematic search is depicted in Figure 3.

### 5.2. Results Based on MARS Rating

Sections A (ENGT) up to section D (INFN) in Figure 4 depict the MARS application quality results. In the Engagement section (Sec A), the addition of notification elements related to activities and target progress tips, modifications of the objectives, and attained goals gave “mySugr” the maximum ENGT mean score of 4.30/5. Meanwhile, “Intellin Diabetes Management” received the lowest ENGT mean score of 2.10/5.

In the Functionality section (Section B), “mySugr” and “BeatO Smart Diabetes Management” with a mean score of 4.00/5 earned the highest rating on functionality. “Diabetes Forum” obtained the lowest FUNT mean score of 2.50/5. The processes for the registration and user login for “Diabetes Forum” required the use of an external glucometer. Thus, these processes were challenging.

“OneTouch Reveal” received the highest score of 4.33/5 in section C for Aesthetics. The application features high-quality visuals and graphic elements, as well as a well-organized and easy interface. The application’s features were further improved by the color design. “Intellin Diabetes Management” received the lowest AEST mean score of 2.5/5.

In the Information section (Section D), the application “mySugr” obtained the highest rating of 3.93/5. The application included a detailed summary of the program’s features as well as clear and attainable goals. The information was well-defined and appropriate for achieving the application’s objectives. The graphic data were straightforward, exact, and easy to understand. The lowest INFN mean score of 2.36/5 was given to the “Diabetes Forum” application. This application did not seem to have any clear, quantifiable objectives.

Table 2 shows the mean of the MARS scores related to the 11 applications downloaded by the two major reviewers. “Alt 2 (mySugr)” was ranked first in Category I on the basis of the mean scores given by both reviewers, with an average application quality mean score of 4.1. With a mean score of 3.87, “Alt 5 (OneTouch Reveal)” came in second. With mean scores of 3.73 and 3.45, the applications “Alt 3 (Diabetes:M)” and “Alt 4 (BeatO Smart Diabetes Management)” placed third and fourth, respectively. With a mean score of 2.42, “Alt 9 (Diabetes Forum)” was the least preferred, thus ranking last. Figure 5 shows the ranking of the various alternatives according to the application quality mean score.

Table 3 displays the results of the subjective application quality assessment. This Table shows that “mySugr” received the maximum Application Subjective Quality Mean Score of 4.5/5 in Category II of MARS, scoring strongly on all categories such as application recommendation, frequency of using the application, willingness of the users to buy the application, and aggregate rating score of the application. The application “Diabetes Forum”, on the other hand, had the lowest score, with a 2.5/5 Application Subjective Quality Mean Score. When compared to the other applications in our survey, the mySugr application provides several sophisticated capabilities. As a result, mySugr came out on top in terms of the AQ mean score (Category I) and the ASQ mean score (Category II). Diabetes Forum was the lowest-ranked application in terms of AQ mean score and ASQ mean score. Figure 6 shows how the different alternatives are ranked depending on the Application Subjective Quality mean score.

For all 11 T2DM mHealth applications assessed in this study, application values based on rating, number of users, mean scores of ENGT, FUNT, AEST, INFN, and application quality, and the application subjective quality mean scores are given in Table 6. As shown in Table 7, the links between user ratings found in application stores, the number of downloads, quality factors specified by the MARS score, and the subjective quality mean scores were investigated using Spearman’s correlation analysis. MARS demonstrated positive relationships between aesthetics and engagement (r = 0.85), ASQ and functionality (r = 0.96), and app-specific items and information (r = 0.98) using Spearman’s correlation analysis.

### 5.3. Results Based on ID3 Algorithm

Initially, since the information gain of Functionality—i.e., 0.74—was found to be the highest, splitting was performed based on the information gain; hence, functionality became the root node of the decision tree. All the examples were negative, i.e., not preferable, when the functionality attribute had a low value. As a result, and as illustrated in Figure 2, we can simply write Not Preferable. Similarly, all of the instances for the high value of the Functionality characteristic are positive, i.e., preferable. However, two examples are positive and one is negative for the moderate value of the Functionality attribute. In this scenario, we cannot just label them as preferable or not preferable. Now, we will move on to the dataset including {A1, A6, A8}. Because Functionality has previously been taken into account, the splitting will now be performed based on the remaining qualities of Engagement, Aesthetics, and Information after determining the Entropy and Information Gain. We then repeat the method until we acquire a decision tree similar to the one illustrated in Figure 7.

## 6. Discussion

There are various ways, such as MCDM methodologies [38,39], MARS, ID3, etc., through which we can determine the usability of T2DM mHealth applications. The purpose of this assessment has been to determine the quality, effectiveness, and functionality of T2DM mHealth applications and to recommend the optimal one for use. To our knowledge, this is the first study to utilize a standardized rating tool called MARS and to develop a decision tree using the ID3 algorithm to perform a systematic review and analyze the effectiveness of mHealth applications that are accessible in the Google Play Store for the avoidance of T2DM among the people of Jharkhand region. We discovered that two applications, “mySugr” and “OneTouch Reveal”, rated well in the Google Play market for the prevention of T2DM among the people of Jharkhand region.

Furthermore, the ID3 algorithm, in contrast to the MARS methodology, helped to develop a decision tree that could be used to predict whether or not an application is preferable in terms of its usability aspect. A1, A2, A3, A4, A6, and A7 were deemed to be preferable among the 11 applications examined, but A5, A8, A9, A10, and A11 were not.

For most populations, lifestyle factors, including lack of proper nutrition, not participating in physical activities, consuming excessive alcohol, cigarette smoking, misuse of drugs, lack of sleep, and mental health stress are well-recognized as key predictors related to metabolic illnesses, including Type 2 Diabetes Mellitus. The majority of these lifestyle habits serve as the foundation for creating the mHealth application.

For its high scores in Engagement, Functionality, and Information, the “mySugr” application ranked well. Meals, food, medications, carbohydrates, and blood glucose levels may all be logged quickly and easily using mySugr. It offers a customizable logging page that allows you to add, remove, and reorganize fields. It can create clever, easy-to-understand blood glucose graphs. Blood glucose levels, A1C, and other metrics can be conveniently analyzed on a daily, weekly, or monthly basis, and reminders can be established. mySugr also helps with tailored goal-setting by providing specific advice and actionable coaching. According to reports on the Google Play store, this application has received over a million downloads since its release.

After mySugr, the “OneTouch Reveal” application came in second place. The Aesthetics score was high for “OneTouch Reveal”. The application has a well-organized and straightforward layout, as well as high-quality graphics and visual design. The application’s features were further improved by the color design.

A mean MARS score of more than 2.75 was found in 9 of the 11 applications examined in this study. This indicates that the effectiveness of these T2DM mHealth applications was satisfactory. When we looked at the specific parts of the application quality mean score, it was discovered that Information received the lowest mean score (3.10) in all the 11 applications, followed by Engagement (3.18). Functionality and Aesthetics had mean scores of 3.22 and 3.23, respectively. This suggests that application engagement and the information offered by the application are areas where it could be improved.

This study may have some limitations related to MARS and the ID3 algorithm. MARS’ internal consistency may be biased because of its format, which requires all items to be scored on a 5-point scale. The specific variance of the item-class could not be managed in the current evaluation because there was no variation in the item format. As a result, the quality factor could be blamed for the item-class variance. Future studies could solve these difficulties by employing an alternative item format. Furthermore, in the case of the ID3 algorithm, if only a limited sample is evaluated, the data may be over-fitted or over-classified. Only one attribute could be checked at a time while making a decision. Continuous data classification can be computationally expensive because multiple trees would have to be constructed to determine where the continuum ends. A slight change in the data can result in a substantial change in the decision tree’s structure, resulting in instability. When compared to other algorithms, decision trees might have significantly more sophisticated calculations. For regression and forecasting continuous values, the decision tree algorithm is insufficient.

## 7. Conclusions

Although a number of lifestyle and behavior change applications for the prevention and management of T2DM are being developed, the scientific database supporting their efficacy is still limited. Two applications, “mySugr” and “OneTouch Reveal”, are highly rated for their usefulness in our review. Further research using randomized controlled trials is required to find the utility of mobile applications in diabetes prevention. Future work may also include the prediction of the best feature using other predictive analysis methodologies such as Random Forest, Logistic Regression, etc. Due to the obvious expansion of current technologies, mobile applications could aid in promoting T2DM self-management. The publicly available T2DM self-management applications reviewed here generally had a variety of functions and capabilities, but the quality was poor, with a lot of room for development. Additional work is required to enhance these applications by incorporating user engagement strategies, providing more detailed and evidence-based information, and supporting a wider range of clinically indicated activities. Extensive scientific reviews of the usefulness and efficacy of these applications for behavioral change and the improvement of health outcomes are also required.

## Figures and Tables

**Figure 1 ijerph-19-06999-f001:**
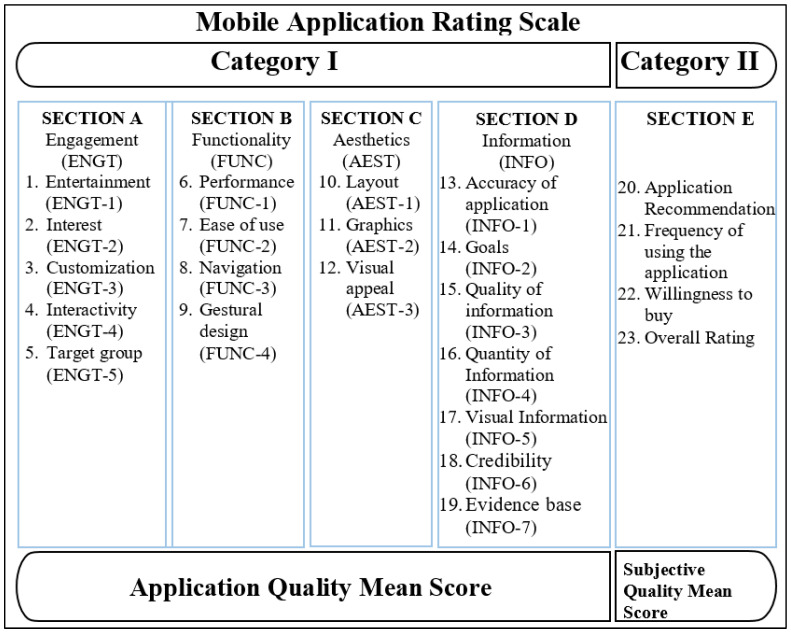
Categories of MARS.

**Figure 2 ijerph-19-06999-f002:**
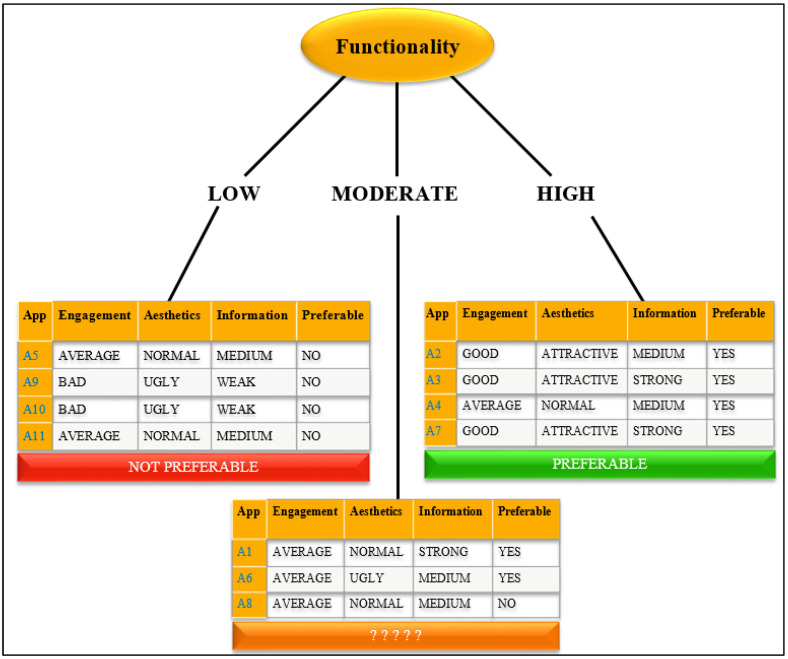
Root node of the ID3 decision tree.

**Figure 3 ijerph-19-06999-f003:**
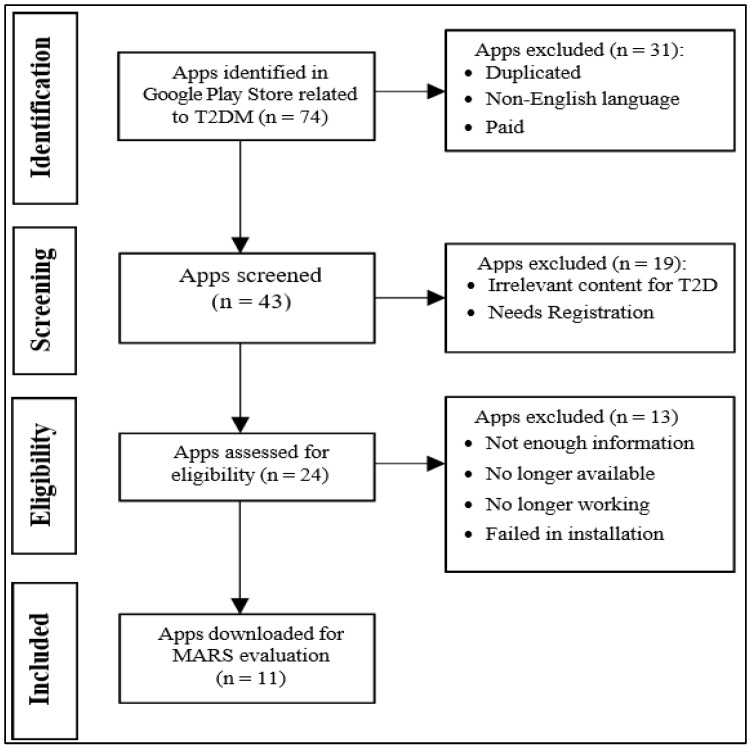
Systematic Search.

**Figure 4 ijerph-19-06999-f004:**
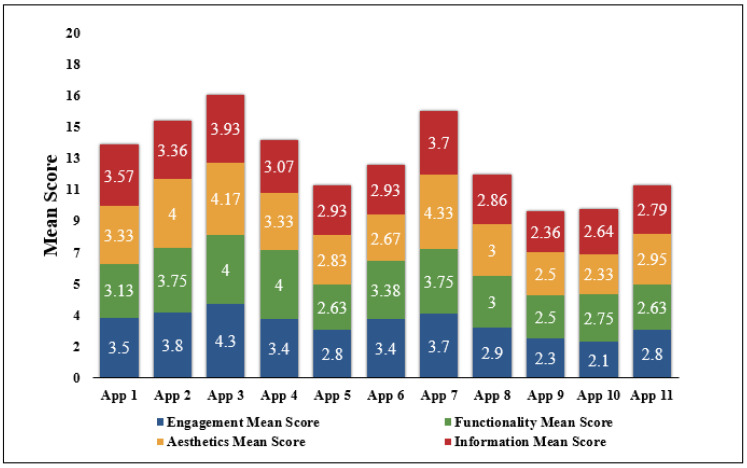
Mean scores associated with MARS Section A (ENGT), Section B (FUNT), Section C (AEST), and Section D (INFN) for all the 11 applications.

**Figure 5 ijerph-19-06999-f005:**
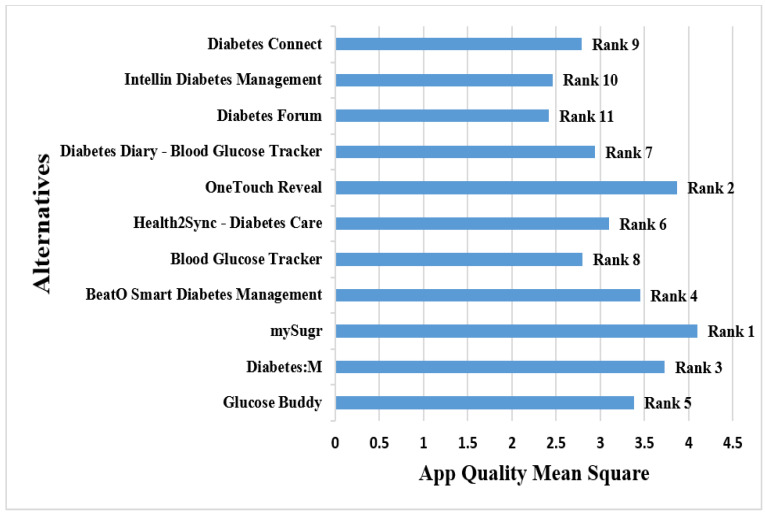
Rank of the applications based on the Application Quality Mean Score.

**Figure 6 ijerph-19-06999-f006:**
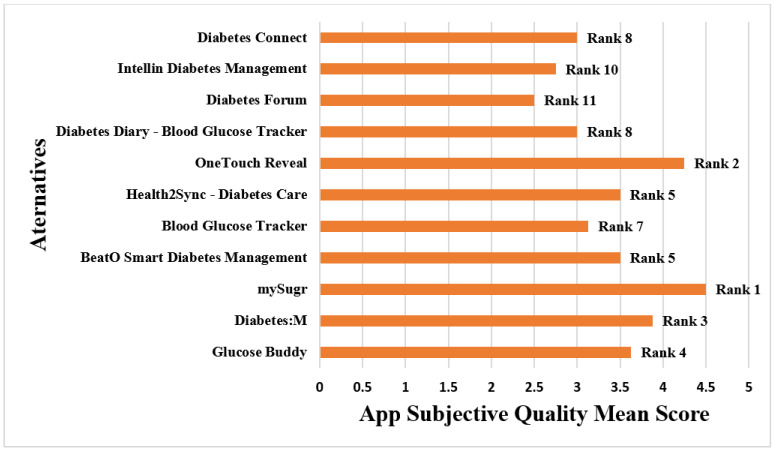
Rank of the applications based on the Application Subjective Quality Mean Score.

**Figure 7 ijerph-19-06999-f007:**
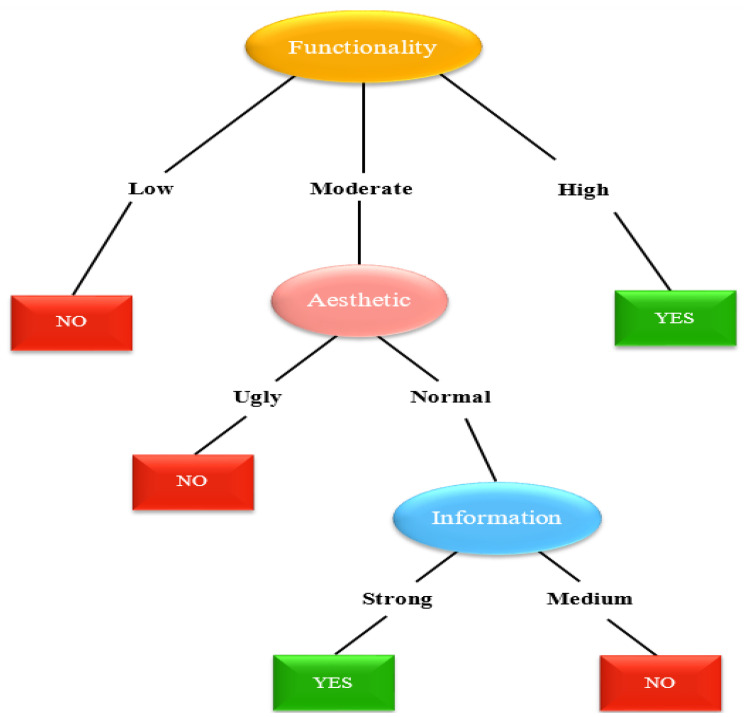
Final Decision Tree.

**Table 1 ijerph-19-06999-t001:** T2DM mHealth apps reviewed in this study.

Application No.	Application Name
App 1	Glucose Buddy
App 2	Diabetes:M
App 3	mySugr
App 4	BeatO Smart Diabetes Management
App 5	Blood Glucose Tracker
App 6	Health2Sync-Diabetes Care
App 7	OneTouch Reveal
App 8	Diabetes Diary-Blood Glucose Tracker
App 9	Diabetes Forum
App 10	Intellin Diabetes Management
App 11	Diabetes Connect

**Table 2 ijerph-19-06999-t002:** Category I, including the Application Quality Mean Score.

Application Name	Attributes	Sub-Attributes	Reviewer 1	Reviewer 2	Mean	Standard Deviation	Application Quality Score
App 1 (Glucose Buddy)	ENGT	ENGT-1	4	3	3.5	0.53	3.38
		ENGT-2	4	3			
		ENGT-3	3	4			
		ENGT-4	3	4			
		ENGT-5	4	3			
	FUNT	FUNT-1	3	3	3.13	0.64	
		FUNT-2	3	4			
		FUNT-3	4	3			
		FUNT-4	2	3			
	AEST	AEST-1	4	4	3.33	0.82	
		AEST-2	4	2			
		AEST-3	3	3			
	INFN	INFN-1	3	4	3.57	0.76	
		INFN-2	4	3			
		INFN-3	3	4			
		INFN-4	4	5			
		INFN-5	4	3			
		INFN-6	3	2			
		INFN-7	4	4			
App 2 (Diabetes:M)	ENGT	ENGT-1	4	4	3.8	0.63	3.73
		ENGT-2	3	3			
		ENGT-3	4	4			
		ENGT-4	4	3			
		ENGT-5	5	4			
	FUNT	FUNT-1	4	3	3.75	0.71	
		FUNT-2	5	4			
		FUNT-3	4	3			
		FUNT-4	3	4			
	AEST	AEST-1	4	4	4.0	0.63	
		AEST-2	4	5			
		AEST-3	3	4			
	INFN	INFN-1	3	5	3.36	0.74	
		INFN-2	4	3			
		INFN-3	3	3			
		INFN-4	3	4			
		INFN-5	4	3			
		INFN-6	3	4			
		INFN-7	2	3			
App 3 (mySugr)	ENGT	ENGT-1	4	5	4.3	0.67	4.1
		ENGT-2	5	4			
		ENGT-3	4	4			
		ENGT-4	5	3			
		ENGT-5	5	4			
	FUNT	FUNT-1	4	4	4	0.76	
		FUNT-2	5	3			
		FUNT-3	4	4			
		FUNT-4	5	3			
	AEST	AEST-1	4	4	4.17	0.41	
		AEST-2	5	4			
		AEST-3	4	4			
	INFN	INFN-1	5	5	3.93	0.73	
		INFN-2	5	4			
		INFN-3	4	4			
		INFN-4	4	3			
		INFN-5	4	4			
		INFN-6	3	3			
		INFN-7	3	4			
App 4 (BeatO Smart Diabetes Management)	ENGT	ENGT-1	4	3	3.4	0.7	3.45
		ENGT-2	3	3			
		ENGT-3	4	4			
		ENGT-4	2	4			
		ENGT-5	3	4			
	FUNT	FUNT-1	4	4	4	0.53	
		FUNT-2	3	4			
		FUNT-3	4	4			
		FUNT-4	5	4			
	AEST	AEST-1	4	3	3.33	0.52	
		AEST-2	4	3			
		AEST-3	3	3			
	INFN	INFN-1	4	3	3.07	0.73	
		INFN-2	3	4			
		INFN-3	3	4			
		INFN-4	3	3			
		INFN-5	2	3			
		INFN-6	2	2			
		INFN-7	3	4			
App 5 (Blood Glucose Tracker)	ENGT	ENGT-1	2	3	2.8	0.42	2.8
		ENGT-2	3	2			
		ENGT-3	3	3			
		ENGT-4	3	3			
		ENGT-5	3	3			
	FUNT	FUNT-1	3	2	2.63	0.74	
		FUNT-2	3	4			
		FUNT-3	2	2			
		FUNT-4	3	2			
	AEST	AEST-1	3	3	2.83	0.41	
		AEST-2	3	2			
		AEST-3	3	3			
	INFN	INFN-1	3	2	2.93	0.62	
		INFN-2	3	3			
		INFN-3	4	3			
		INFN-4	3	4			
		INFN-5	3	2			
		INFN-6	2	3			
		INFN-7	3	3			
App 6 (Health2Sync-Diabetes Care)	ENGT	ENGT-1	4	5	3.4	0.84	3.1
		ENGT-2	4	3			
		ENGT-3	2	3			
		ENGT-4	3	4			
		ENGT-5	3	3			
	FUNT	FUNT-1	3	4	3.38	0.74	
		FUNT-2	4	3			
		FUNT-3	3	4			
		FUNT-4	2	4			
	AEST	AEST-1	3	3	2.67	0.52	
		AEST-2	2	2			
		AEST-3	3	3			
	INFN	INFN-1	2	3	2.93	0.73	
		INFN-2	3	3			
		INFN-3	3	4			
		INFN-4	3	4			
		INFN-5	2	4			
		INFN-6	2	3			
		INFN-7	2	3			
App 7 (OneTouch Reveal)	ENGT	ENGT-1	4	4	3.7	0.82	3.87
		ENGT-2	5	5			
		ENGT-3	4	3			
		ENGT-4	3	3			
		ENGT-5	3	3			
	FUNT	FUNT-1	4	5	3.75	0.71	
		FUNT-2	4	3			
		FUNT-3	3	4			
		FUNT-4	3	4			
	AEST	AEST-1	4	5	4.33	0.82	
		AEST-2	5	4			
		AEST-3	3	5			
	INFN	INFN-1	3	4	3.7	0.61	
		INFN-2	3	5			
		INFN-3	4	4			
		INFN-4	3	4			
		INFN-5	4	3			
		INFN-6	3	4			
		INFN-7	4	4			
App 8 (Diabetes Diary-Blood Glucose Tracker)	ENGT	ENGT-1	3	4	2.9	0.74	2.94
		ENGT-2	3	2			
		ENGT-3	3	2			
		ENGT-4	3	3			
		ENGT-5	4	2			
	FUNT	FUNT-1	3	3	3	0.53	
		FUNT-2	3	4			
		FUNT-3	3	3			
		FUNT-4	2	3			
	AEST	AEST-1	2	3	3	0.63	
		AEST-2	3	4			
		AEST-3	3	3			
	INFN	INFN-1	4	3	2.86	0.66	
		INFN-2	4	3			
		INFN-3	3	3			
		INFN-4	2	3			
		INFN-5	3	3			
		INFN-6	2	2			
		INFN-7	2	3			
App 9 (Diabetes Forum)	ENGT	ENGT-1	2	3	2.3	0.67	2.42
		ENGT-2	3	3			
		ENGT-3	2	2			
		ENGT-4	2	1			
		ENGT-5	2	3			
	FUNT	FUNT-1	2	3	2.5	0.53	
		FUNT-2	2	2			
		FUNT-3	3	2			
		FUNT-4	3	3			
	AEST	AEST-1	2	3	2.5	0.55	
		AEST-2	3	2			
		AEST-3	2	3			
	INFN	INFN-1	3	2	2.36	0.63	
		INFN-2	2	3			
		INFN-3	3	3			
		INFN-4	2	3			
		INFN-5	3	2			
		INFN-6	2	1			
		INFN-7	2	2			
App 10 (Intellin Diabetes Management)	ENGT	ENGT-1	2	3	2.1	0.57	2.46
		ENGT-2	2	2			
		ENGT-3	1	2			
		ENGT-4	3	2			
		ENGT-5	2	2			
	FUNT	FUNT-1	3	3	2.75	0.46	
		FUNT-2	2	3			
		FUNT-3	3	2			
		FUNT-4	3	3			
	AEST	AEST-1	2	3	2.33	0.52	
		AEST-2	2	2			
		AEST-3	2	3			
	INFN	INFN-1	3	2	2.64	0.5	
		INFN-2	3	2			
		INFN-3	3	3			
		INFN-4	2	3			
		INFN-5	3	2			
		INFN-6	3	2			
		INFN-7	3	3			
App 11 (Diabetes Connect)	ENGT	ENGT-1	3	2	2.8	0.63	2.79
		ENGT-2	3	4			
		ENGT-3	2	3			
		ENGT-4	2	3			
		ENGT-5	3	3			
	FUNT	FUNT-1	2	3	2.63	0.52	
		FUNT-2	3	3			
		FUNT-3	2	3			
		FUNT-4	3	2			
	AEST	AEST-1	3	4	2.95	0.52	
		AEST-2	3	4			
		AEST-3	3	3			
	INFN	INFN-1	3	2	2.79	0.7	
		INFN-2	3	3			
		INFN-3	2	2			
		INFN-4	3	4			
		INFN-5	3	4			
		INFN-6	2	3			
		INFN-7	2	3			

**Table 3 ijerph-19-06999-t003:** Category II, including the application’s subjective quality mean score.

Application Name	Parameters	Reviewer 1	Reviewer 2	Mean Score
App 1 (Glucose Buddy)	App Recommendation	3	5	3.63
	Frequency of using the app	4	4	
	Willingness to buy	4	3	
	Overall Rating	3	3	
App 2 (Diabetes:M)	App Recommendation	3	4	3.88
	Frequency of using the app	4	4	
	Willingness to buy	4	3	
	Overall Rating	5	4	
App 3 (mySugr)	App Recommendation	4	5	4.5
	Frequency of using the app	5	4	
	Willingness to buy	4	5	
	Overall Rating	4	5	
App 4 (BeatO Smart Diabetes Management)	App Recommendation	4	3	3.5
	Frequency of using the app	4	3	
	Willingness to buy	4	3	
	Overall Rating	4	3	
App 5 (Blood Glucose Tracker)	App Recommendation	3	3	3.13
	Frequency of using the app	3	3	
	Willingness to buy	3	4	
	Overall Rating	3	3	
App 6 (Health2Sync-Diabetes Care)	App Recommendation	3	4	3.5
	Frequency of using the app	3	4	
	Willingness to buy	4	3	
	Overall Rating	3	4	
App 7 (OneTouch Reveal)	App Recommendation	4	5	4.25
	Frequency of using the app	5	4	
	Willingness to buy	4	4	
	Overall Rating	4	4	
App 8 (Diabetes Diary-Blood Glucose Tracker)	App Recommendation	3	4	3
	Frequency of using the app	3	4	
	Willingness to buy	2	3	
	Overall Rating	3	2	
App 9 (Diabetes Forum)	App Recommendation	2	3	2.5
	Frequency of using the app	3	2	
	Willingness to buy	2	4	
	Overall Rating	2	2	
App 10 (Intellin Diabetes Management)	App Recommendation	4	3	2.75
	Frequency of using the app	2	3	
	Willingness to buy	3	2	
	Overall Rating	3	2	
App 11 (Diabetes Connect)	App Recommendation	3	3	3
	Frequency of using the app	4	3	
	Willingness to buy	3	2	
	Overall Rating	3	3	

**Table 4 ijerph-19-06999-t004:** Values assigned to attributes based on the mean score.

Attributes	Values	Mean Score Range
Engagement (ENGT)	Bad	≤2.75
	Average	>2.75 & ≤3.5
	Good	>3.5
Functionality (FUNT)	Low	≤2.75
	Moderate	>2.75 & ≤3.5
	High	>3.5
Aesthetics (AEST)	Ugly	≤2.75
	Normal	>2.75 & ≤3.5
	Attractive	>3.5
Information (INFN)	Weak	≤2.75
	Medium	>2.75 & ≤3.5
	Strong	>3.5

**Table 5 ijerph-19-06999-t005:** Decision-making factors to select the application (data set).

Application	Engagement	Functionality	Aesthetics	Information	Application Selection
A1	Average	Moderate	Normal	Strong	YES
A2	Good	High	Attractive	Medium	YES
A3	Good	High	Attractive	Strong	YES
A4	Average	High	Normal	Medium	YES
A5	Average	Low	Normal	Medium	NO
A6	Average	Moderate	Ugly	Medium	YES
A7	Good	High	Attractive	Strong	YES
A8	Average	Moderate	Normal	Medium	NO
A9	Bad	Low	Ugly	Weak	NO
A10	Bad	Low	Ugly	Weak	NO
A11	Average	Low	Normal	Medium	NO

**Table 6 ijerph-19-06999-t006:** Application values based on rating, number of users, mean score of ENGT, FUNT, AEST, INFN, and AQ, and the ASQ Mean Scores.

	App 1	App 2	App 3	App 4	App 5	App 6	App 7	App 8	App 9	App 10	App 11
**Ratings in Google Play Store**	4.4	4.3	4.3	4.2	4.5	4.7	3.8	4.5	4.4	3.9	4.2
**Number of users**	14,540	21,475	63,207	19,050	19,233	13,050	29,436	2101	888	555	4768
**Engagement**	3.5	3.8	4.3	3.4	2.8	3.4	3.7	2.9	2.3	2.1	2.8
**Functionality**	3.13	3.75	4	4	2.63	3.38	3.75	3	2.5	2.75	2.63
**Aesthetics**	3.33	4	4.17	3.33	2.83	2.67	4.33	3	2.5	2.33	2.95
**Information**	3.57	3.36	3.93	3.07	2.93	2.93	3.7	2.86	2.36	2.64	2.79
**App Subjective Quality Mean Score**	3.63	3.88	4.5	3.5	3.13	3.5	4.25	3	2.5	2.75	3
**App Quality Mean Score**	3.38	3.73	4.1	3.45	2.8	3.1	3.87	2.94	2.42	2.46	2.79

**Table 7 ijerph-19-06999-t007:** Correlation between the major specifications of applications and the MARS domains.

	Ratings in Google Play Store	Number of Users	Engagement	Functionality	Aesthetics	Information	Application Subjective Quality Mean Score	Application Quality Mean Square
**Ratings in Google Play Store**	1							
**Number of users**	−0.097	1						
**Engagement**	0.05	0.83	1					
**Functionality**	−0.18	0.72	0.87	1				
**Aesthetics**	−0.33	0.78	0.88	0.79	1			
**Information**	−0.21	0.84	0.91	0.76	0.90	1		
**Application Subjective Quality Mean Score**	−0.18	0.88	0.96	0.86	0.91	0.96	1	
**Application Quality Mean Square**	−0.17	0.84	0.97	0.91	0.95	0.94	0.98	1

## Data Availability

Data Sharing not applicable.

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
