# Peer review of "Usability Evaluation and Classification of mHealth Applications for Type 2 Diabetes Mellitus Using MARS and ID3 Algorithm"

_ijerph, 2022, doi:10.3390/ijerph19126999_

Round 1
Reviewer 1 Report
Many chronic diseases, such as diabetes, can be treated with the help of smart devices and mHealth applications. Patients monitor their health data autonomously and thus relieve their doctors, who can focus on emergency cases. However, it is crucial to find the best health applications from the vast number of available ones. For this purpose, the authors analyze in their paper entitled "Usability Evaluation and Classification of mHealth Applications for Type 2 Diabetes Mellitus using MARS and ID3 algorithm" mHealth applications for patients suffering from diabetes. After a pre-selection, the authors take a closer look at the eleven remaining applications using the MARS method. Then, based on the calculated scores, the authors train a decision tree that is intended to determine whether a certain mHealth application is useful or not.
While such an assessment is immensely helpful for patients, and by means of a decision tree the usefulness of new applications can be (theoretically) determined, I have major concerns about the approach and the presentation of the results.
First of all, the manuscript seems to be very unstructured. For example, Section 2.1 discusses related work on MARS and Section 2.2 discusses related work on the ID3 algorithm, but explanations regarding MARS, decision trees, and the ID3 algorithm are provided in Section 3. Also, the discussion provided in Section 2.2 does not appear to be relevant. The fact that decision trees can be used to solve classification problems is most likely out of question. Therefore, what insight does the reader gain by listing these research papers? Can any conclusions be drawn from them for the paper at hand? Information regarding the selection of the analyzed applications is only given at the very end on page 18.
The authors use several exclusion criteria when selecting the applications under review. However, these are not explained. For example, why are only free applications considered? If the goal is to find the best diabetes application, why does the price matter? Such restrictions result in the very small number of eleven diabetes applications remaining for investigation. That seems negligible considering the large number of such health applications available in app stores. But especially since so few applications are considered, I would have liked to see a brief introduction of the applications. Apart from mentioning the names of the applications, the reader does not receive any information about them. Therefore, the ratings cannot be verified. Also, there are no details given about the reviewers. Do they have domain knowledge so that their ratings are really sound? Since no information is given on how the evaluation was actually carried out, the scores given in Table 2 and Table 3 (i.e., pages 9 to 14!) are not comprehensible and have little meaning to the reader. Also, some of the evaluation criteria are questionable. For example, it is strange that the reviewers evaluate whether they would be willing to buy the application - since only free applications are analyzed, this issue seems to be irrelevant in this context.
While in the context of the MARS analysis, I am primarily missing information that I believe the authors should provide, I question the following classification per se. First of all, the authors considerably reduce their data basis from the MARS analysis by mapping the continuous scores of the four main categories engagement, functionality, aesthetics, and information to one of three discrete values. The threshold values chosen here are not discussed. Since other threshold values would completely change the result, it is mandatory to communicate them and not to regard them as self-explanatory. The authors add a fifth attribute to these four attributes, which indicates whether the application is recommended. This attribute is also calculated from the MARS scores and then mapped to the discrete values yes and no. This last attribute represents the label for the classification. To train the decision tree, the authors rely on the eleven applications. That is, of the eighty-one possible combinations that the first four attributes can have, the authors consider eleven at most. I would doubt that this is sufficient for training the decision tree. Also, the purpose is not clear to me. In the best case, the decision tree would learn the threshold applied by the authors to calculate the label. In the worst case, it is learning non-existent patterns, which are caused by a too small data base. Nevertheless, for any new application, the MARS analysis would still have to be performed first. So, what benefit does the tree provide? Since the authors have no test data, they cannot determine the accuracy of their tree (or any other relevant metrics). Also, the lengthy description of the manual calculation of the split criteria does not provide any benefit to the reader.
While I see added value for the reader in the descriptive part of the analysis - but then comprehensive additional explanations of the applications, how the scoring is performed, etc. are needed - I see little to no benefit in this type of prediction. Of course, a decision tree can be trained, but the informative value is very questionable, and the applicability is equal to zero. To use the decision tree, the MARS analysis would have to be carried out first for each new application. In the course of this analysis, the label for the new application is determined anyway. Thus, the second contribution "Choosing the best mHeath applications on T2DM using ID3 decision making algorithm" is not provided.
Furthermore, there are several typos, formatting issues, and referencing errors in the paper. For instance
- Line 85 "table 1"
- Line 86 "Indians.Table"
- Line 211 "10thand 12thboard"
- Line 396 "table 1" (it should refer to Table 5).
- etc.
Author Response
Comment 1.
First of all, the manuscript seems to be very unstructured. For example, Section 2.1 discusses related work on MARS and Section 2.2 discusses related work on the ID3 algorithm, but explanations regarding MARS, decision trees, and the ID3 algorithm are provided in Section 3. Also, the discussion provided in Section 2.2 does not appear to be relevant.
Response: Changes have been made in the manuscript to present it in a more structured way. Section 2 has been made Section 3 and Section 3 has been made Section 2.
Section 2.2 which is presently listed as 3.2 highlights the importance of using the Classification Decision Tree i.e. ID3 algorithm. The reason behind choosing ID3 is that ID3 algorithm builds a decision tree using a top-down greedy technique which searches the given sets of training data for testing the attribute at each node and generates comprehensible predictions. The main issue with any decision tree algorithm is choosing the splitting attribute, which is done efficiently by ID3 algorithm based on Entropy Metric and Information Gain. The attribute with the highest information gain is selected as the splitting attribute. Also in ID3, if the final attribute is identified it will prune the test data sets thus reducing the test counts.
Comment 2.
The fact that decision trees can be used to solve classification problems is most likely out of question. Therefore, what insight does the reader gain by listing these research papers? Can any conclusions be drawn from them for the paper at hand? Information regarding the selection of the analyzed applications is only given at the very end on page 18.
Response: A decision tree that can be used for solving classification problems is a known fact as mentioned by the reviewer. But there are various classification algorithms available to build decision trees such as ID3, C4.5, CART, CHAID etc. However, in this research work, we have listed various research papers which work on classification problems, especially on the ID3 (Iterative Dichotomiser) algorithm.
The reason behind listing these research papers in the present work is to focus on how ID3 algorithm helps in the prediction of best mHealth applications on T2DM. ID3 algorithm builds decision tree using a top-down greedy technique which searches the given sets of training data for testing the attribute at each node and generating comprehensible predictions. The main issue with any decision tree algorithm is choosing the splitting attribute, which is done efficiently by ID3 algorithm based on Entropy Metric and Information Gain. The attribute with the highest information gain is selected as the splitting attribute. Also in ID3, if the final attribute is identified it will prune the test data sets thus reducing the test counts.
Thus the research papers highlighted the benefits of using ID3 algorithm for analyzing the mHealth applications in the proposed work.
Comment 3.
The authors use several exclusion criteria when selecting the applications under review. However, these are not explained.
Response: Explanations have been given below and highlighted in Section 4.1 of the revised manuscript.
Paid applications, non-English language applications, non-T2DM specific applications, and duplicated applications were all excluded from the Identification phase employing PRISMA. Exclusion criteria throughout the screening phase were irrelevant content for T2DM and the need for registration. The exclusion criteria for the Eligibility phase were: insufficient information, and no longer working. The rest of the applications were downloaded and analyzed with MARS during the Included phase, and a decision tree was created with the help of the ID3 algorithm.
Comment 4.
Why are only free applications considered? If the goal is to find the best diabetes application, why does the price matter? Such restrictions result in the very small number of eleven diabetes applications remaining for investigation. That seems negligible considering the large number of such health applications available in app stores. But especially since so few applications are considered, I would have liked to see a brief introduction of the applications. Apart from mentioning the names of the applications, the reader does not receive any information about them. Therefore, the ratings cannot be verified.
Response: We will consider the paid applications in the future scope of the study. A brief explanation of each application used in our study is given in section 2.1
Comment 5.
Also, there are no details given about the reviewers. Do they have domain knowledge so that their ratings are really sound?
Response: The two reviewers selected in our study are having the following domain knowledge, which is also reflected under Section 4.1 “Quality Assessment of the applications using MAR”:
- One is having a master’s degree in health information management and is presently working as a research consultant in a medical firm, especially on diabetes.
- Another reviewer is having master’s degree in medical informatics and is working as a dietitian for several years.
Comment 6.
Since no information is given on how the evaluation was actually carried out, the scores given in Table 2 and Table 3 (i.e., pages 9 to 14!) are not comprehensible and have little meaning to the reader. Also, some of the evaluation criteria are questionable. For example, it is strange that the reviewers evaluate whether they would be willing to buy the application - since only free applications are analyzed, this issue seems to be irrelevant in this context.
Response: The MARS is a multidimensional instrument assessing the quality of mHealth applications. The quality assessment is done using standard Questionnaires which consist of a total of 19 items covering four dimensions. The dimensions are (A) engagement (5 items: fun, interest, individual adaptability, interactivity, target group), (B) functionality (4 items: performance, usability, navigation, gestural design), (C) aesthetics (3 items: layout, graphics, visual appeal), and (D) information quality (7 items: accuracy of app description, goals, quality of information, the quantity of information, quality of visual information, credibility, evidence base). All items are assessed on a 5-point scale (1-inadequate, 2-poor, 3-acceptable, 4-good, and 5-excellent). Items assessing information quality can also be rated as not applicable (e.g., in case of missing evidence or missing visual information).
The applications which we have analyzed is a free version. However, they are also have paid/ premium version of the mHealth applications which comes with additional or premium features. In this research work, we tried to get an idea about the usability aspect of each application. In future work, paid mHealth applications may be considered for the study.
Comment 7.
While in the context of the MARS analysis, I am primarily missing information that I believe the authors should provide, I question the following classification per se. First of all, the authors considerably reduce their data basis from the MARS analysis by mapping the continuous scores of the four main categories engagement, functionality, aesthetics, and information to one of three discrete values. The threshold values chosen here are not discussed. Since other threshold values would completely change the result, it is mandatory to communicate them and not to regard them as self-explanatory.
Response: The concept of threshold value has been removed as it is mentioned earlier by mistake.
Comment 8.
The authors add a fifth attribute to these four attributes, which indicates whether the application is recommended. This attribute is also calculated from the MARS scores and then mapped to the discrete values yes and no. This last attribute represents the label for the classification. To train the decision tree, the authors rely on the eleven applications. That is, of the eighty-one possible combinations that the first four attributes can have, the authors consider eleven at most. I would doubt that this is sufficient for training the decision tree. Also, the purpose is not clear to me. In the best case, the decision tree would learn the threshold applied by the authors to calculate the label. In the worst case, it is learning non-existent patterns, which are caused by a too-small database. Nevertheless, for any new application, the MARS analysis would still have to be performed first. So, what benefit does the tree provide? Since the authors have no test data, they cannot determine the accuracy of their tree (or any other relevant metrics). Also, the lengthy description of the manual calculation of the split criteria does not provide any benefit to the reader.
Response: In our proposed work we are going to provide the users with the best mHealth applications for self-monitoring of T2DM patients. For doing these we have classified the mHealth applications using two methods. One is MARS which provides a ranking of the mHealth applications and has the outcome of the analysis is continuous in nature i.e. it provides an Application Quality Score based on four attributes Engagement, Functionality, Aesthetics and Information. Also, it provides a ranking of the mHealth applications based on the Subjective Quality Mean Score. Basically, it uses a regression decision tree for calculating the score based on which the ranking is done.
But if we want to provide the users with a decision tree where they can classify the mHealth applications into two cases- recommended or not recommended then MARS method is not sufficient. For this, we use ID3 algorithm which uses a classification decision tree using a top-down greedy search approach. As we know the greedy algorithm always makes the choice that seems to be the best at that moment. It calculates Entropy and Information Gain of every splitting attribute and based on the result it categorizes the mHealth applications on two categories- recommended or not recommended.
Thus, with two types of decision tree algorithms, we want to help the users to identify the best mHealth applications for self-monitoring of T2DM.
And regarding the training set, yes at this point we are having very less training set. In future, we will work on large training data set which will help in the better recommendation of mHealth applications.
Comment 9.
While I see added value for the reader in the descriptive part of the analysis - but then comprehensive additional explanations of the applications, how the scoring is performed, etc. are needed - I see little to no benefit in this type of prediction. Of course, a decision tree can be trained, but the informative value is very questionable, and the applicability is equal to zero. To use the decision tree, the MARS analysis would have to be carried out first for each new application. In the course of this analysis, the label for the new application is determined anyway. Thus, the second contribution "Choosing the best mHeath applications on T2DM using ID3 decision making algorithm" is not provided.
Response: If we want to provide the users with a decision tree where they can classify the mHealth applications into two cases- recommended or not recommended then MARS method is not sufficient. In contrast to MARS, ID3 (Iterative Dichotomiser 3) is a classification technique that generates a decision tree from a dataset. The ID3 algorithm starts with the root node, which is the original set. The algorithm iterates through every unused attribute in the set and estimates the entropy or information gain of that attribute on each iteration. The property with the lowest entropy (or greatest information gain) is then chosen. To produce subsets of the data, the set is split or partitioned by the selected attribute. The process recurses on each subset, considering only qualities that have never been chosen previously.
The decision tree is formed throughout the procedure, with each non-terminal node (internal node) reflecting the attribute upon which data was split and terminal nodes (leaf nodes) providing the class label of the final subset related to this branch.
Comment 10.
Furthermore, there are several typos, formatting issues, and referencing errors in the paper. For instance
Line 85 "table 1"
Line 86 "Indians.Table"
Line 211 "10thand 12thboard"
Line 396 "table 1" (it should refer to Table 5).
etc.
Response: All the typing errors, formatting issues and referencing errors have been resolved in the revised manuscript.
Reviewer 2 Report
This work presented an usability evaluation and classification of mHealth Applications for Type 2 Diabetes Mellitus using MARS and ID3 algorithm. It is suggested to make the following settings:
- Add a reviewer in the quality evaluation of applications using MARS.
- Specify the limitations of the work when using MARS and the ID3 algorithm.
- Detail the description of the use of the selected applications by the evaluators.
Author Response
Comment 1.
Add a reviewer in the quality evaluation of applications using MARS.
Response: Two reviewers along with their domain knowledge and other details have been added to the revised manuscript for quality evaluation of mHealth applications.
Comment 2.
Specify the limitations of the work when using MARS and the ID3 algorithm.
Response: The limitations of the work when using MARS and the ID3 algorithm have been given specified in section 6 i.e. Discussion section
Comment 3.
Detail the description of the use of the selected applications by the evaluators
Response: A brief description of the use of the selected applications by the evaluators has been given in Section 2.1
Round 2
Reviewer 1 Report
First of all, I have to point out that the authors have done a decent job regarding the revision of their paper, and this has significantly increased its quality. Especially my comments about the descriptive analysis parts, i.e., the MARS approach, have been addressed comprehensively and my concerns in this respect have been resolved.
What I still question, however, is the predictive analysis using the ID3 algorithm. Perhaps my comments in my review were unclear. I do not question that a decision tree can be trained using ID3, nor that a classification can be performed using a decision tree. The question is rather, is the methodology sound and are the results reliable. Unfortunately, I do not think that either of these is the case here. First of all, I do not see any reason to use ID3 in this use case. Especially when dealing with small data (and in the field of data science, base data with far more than eleven samples are considered as small data), ensemble techniques, such as random forest, are much better suited. Single models tend to overfit very easily for small data, i.e., the tiny amount of base data is described perfectly by the model, but the model hardly reflects reality. In other words, phantom patterns are detected in the data. One could also train a model for four other attributes that most certainly have no impact on deciding whether an app is worth recommending, e.g., shoe size of the app developer. If the base data are small enough, this also leads to a decision tree that perfectly describes the samples – yet the shoe size still does not affect the app quality. Whether overfitting or training on phantom patterns has taken place can only be seen if test data are also available. Only then is it possible to evaluate the quality of the model, i.e., how accurately it represents reality. Without such an evaluation, the decision tree has no significance. That is, using eleven samples as base data cannot enable to make sound predictions.
If the predictive analysis part is still important to the authors, then it needs to be revised from my point of view. First of all, an explanation has to be given why the usage of a decision tree is reasonable instead of a usually much more accurate ensemble technique. This could be due to the fact that a decision tree is easier to interpret by humans. That is, if the goal is to show which attributes of an app determine whether it is recommendable or not, then the decision tree is better suited. But it is not mentioned in the paper whether this is the goal. If I only want to make a decision, I see no reason not to use a method with a higher accuracy. At this point, I would also avoid a discussion about which algorithm is best to use to train the decision tree. If doing so, one would have to provide reasons, why not use for instance its successor C4.5. If, on the other hand, the choice of the algorithm is not important, then a detailed discussion of ID3 in particular is not necessary. The statement that other research works used ID3 for other decision tasks is not applicable as a reason in my opinion – as I said in the beginning, it is out of question that ID3 generates a decision tree, yet so do C4.5, CART, … as well.
Furthermore, I do not think the authors can present the tree itself as a result. Due to the far too small amount of data and the lack of verification by test data, the tree as described above is not reliable. At most, one could describe the procedure to indicate how one could perform a predictive analysis if there was a sufficient amount of base data. Here, however, the procedure must be the contribution, not the resulting decision tree. If the authors intend to do this, however, I would expect that topics such as data wrangling, feature selection, or feature engineering are also addressed up front. Currently, only the features that have been somehow selected and prepared are presented, not the steps that have been taken to get there. In addition, it is crucial to point out the split into training and test data, as this is part of sound supervised learning. Then it would be the contribution for readers, that they – provided appropriate base data are available – would be able to train a classifier by themselves. In my opinion, the decision tree itself is not suitable as a contribution in its current form due to the inappropriate amount of base data and the missing evaluation.
Author Response
First of all, I have to point out that the authors have done a decent job regarding the revision of their paper, and this has significantly increased its quality. Especially my comments about the descriptive analysis parts, i.e., the MARS approach, have been addressed comprehensively and my concerns in this respect have been resolved.
Response: Thank you for your valuable suggestions and comments.
What I still question, however, is the predictive analysis using the ID3 algorithm. Perhaps my comments in my review were unclear. I do not question that a decision tree can be trained using ID3, nor that a classification can be performed using a decision tree. The question is rather, is the methodology sound and are the results reliable. Unfortunately, I do not think that either of these is the case here. First of all, I do not see any reason to use ID3 in this use case. Especially when dealing with small data (and in the field of data science, base data with far more than eleven samples are considered as small data), ensemble techniques, such as random forest, are much better suited. Single models tend to overfit very easily for small data, i.e., the tiny amount of base data is described perfectly by the model, but the model hardly reflects reality. In other words, phantom patterns are detected in the data. One could also train a model for four other attributes that most certainly have no impact on deciding whether an app is worth recommending, e.g., shoe size of the app developer. If the base data are small enough, this also leads to a decision tree that perfectly describes the samples – yet the shoe size still does not affect the app quality. Whether overfitting or training on phantom patterns has taken place can only be seen if test data are also available. Only then is it possible to evaluate the quality of the model, i.e., how accurately it represents reality. Without such an evaluation, the decision tree has no significance. That is, using eleven samples as base data cannot enable to make sound predictions.
Response: I do agree with the learned reviewer regarding dealing with small data set. However, we have observed from various literature review papers which work on ID3 algorithm that the model also performs well for the small dataset. This is our future scope of the study. For the time being while implementing the proposed ID3 model, we have tested with some random data sets, which shows that the model performs accurately. Hence, I would request the reviewer to consider the proposed model with ID3 and in future, we will be implementing advanced ID3 techniques as well ensemble techniques like the random forest. Also in future, we will be considering more samples for training as well as testing.
If the predictive analysis part is still important to the authors, then it needs to be revised from my point of view. First of all, an explanation has to be given why the usage of a decision tree is reasonable instead of a usually much more accurate ensemble technique. This could be due to the fact that a decision tree is easier to interpret by humans. That is, if the goal is to show which attributes of an app determine whether it is recommendable or not, then the decision tree is better suited. But it is not mentioned in the paper whether this is the goal. If I only want to make a decision, I see no reason not to use a method with a higher accuracy. At this point, I would also avoid a discussion about which algorithm is best to use to train the decision tree. If doing so, one would have to provide reasons, why not use for instance its successor C4.5. If, on the other hand, the choice of the algorithm is not important, then a detailed discussion of ID3 in particular is not necessary. The statement that other research works used ID3 for other decision tasks is not applicable as a reason in my opinion – as I said in the beginning, it is out of question that ID3 generates a decision tree, yet so do C4.5, CART, … as well.
Response: Initially, since the information gain of Functionality i.e. 0.74 is found to be the highest, splitting was done based on the information gain and hence, functionality becomes the root node of the decision tree. All the examples are negative, i.e. not preferable, when the functionality attribute has a low value. As a result, we can simply write Not Preferable. Similarly, all of the instances for the high value of the functionality characteristic are positive, i.e. preferable. However, two examples are positive and one is negative for the moderate value of the functionality attribute. In this scenario, we can't just put preferably or not preferable. So now we'll move on to the dataset including {A1, A6, A8}. Because Functionality has previously been taken into account, now the splitting will be done based on the remaining qualities such as Engagement, Aesthetics, and Information after determining the Entropy and Information Gain. Now, among these attributes, Aesthetics has the highest information again and hence splitting will now be done based on Aesthetics. We repeat the method until we acquire a decision tree.
Furthermore, we have also used C 4.5 algorithm for constructing the decision tree for the same attributes. In C 4.5 algorithm, splitting was done based on the Gain Ratio which is calculated by dividing the Information gain by split info. In the case of C 4.5 too, the attribute Functionality has the highest Gain Ratio and hence it becomes the root node of the decision attribute. The procedure is repeated and we obtained a decision tree which is the same as generated using the ID3 algorithm.
Furthermore, I do not think the authors can present the tree itself as a result. Due to the far too small amount of data and the lack of verification by test data, the tree as described above is not reliable. At most, one could describe the procedure to indicate how one could perform a predictive analysis if there was a sufficient amount of base data. Here, however, the procedure must be the contribution, not the resulting decision tree. If the authors intend to do this, however, I would expect that topics such as data wrangling, feature selection, or feature engineering are also addressed upfront. Currently, only the features that have been somehow selected and prepared are presented, not the steps that have been taken to get there. In addition, it is crucial to point out the split into training and test data, as this is part of sound supervised learning. Then it would be the contribution for readers, that they – provided appropriate base data are available – would be able to train a classifier by themselves. In my opinion, the decision tree itself is not suitable as a contribution in its current form due to the inappropriate amount of base data and the missing evaluation.
Response: Features selected in our study are based on the MARS attributes, which are used for representing the decision tree. These attributes are taken specifically for evaluating Mobile App Rating Scale (MARS). However, the features are mentioned in Figure 1. While implementing in python we have divided our data set into a training set(75%) and testing set(25%) which results in 8 rows for training and 3 samples for testing. However, we have also calculated the accuracy of the model which appears to be 0.91. In the future scope of our study, we will implement it with large data sets.